# Organic carbon mass accumulation rate regulates the flux of reduced substances from the sediments of deep lakes

Thomas Steinsberger[1,2], Martin Schmid[1], Alfred Wüest[1,3], Robert Schwefel[3], Bernhard Wehrli[1,2], Beat Müller[1]

[1] Eawag, Swiss Federal Institute of Aquatic Science and Technology, CH-6047 Kastanienbaum, Switzerland

[2] Institute of Biogeochemistry and Pollutant Dynamics, ETH Zurich, CH-8092 Zurich, Switzerland

[3] Physics of Aquatic Systems Laboratory, Margaretha Kamprad Chair, École Polytechnique Fédérale de Lausanne, Institute of Environmental Engineering, CH-1015 Lausanne, Switzerland.

*Correspondence to:* Beat Müller (Beat.Mueller@eawag.ch)

**Abstract.** The flux of reduced substances, such as methane and ammonium, from the sediment to the bottom water ($F_{red}$) is one of the major factors contributing to the consumption of oxygen in the hypolimnia of lakes and thus crucial for lake oxygen management. This study presents fluxes based on sediment porewater measurements from different water depths of five deep lakes of differing trophic states. In meso- to eutrophic lakes $F_{red}$ was directly proportional to the total organic carbon mass accumulation rate (TOC-MAR) of the sediments. TOC-MAR and thus $F_{red}$ in eutrophic lakes decreased systematically with increasing mean hypolimnion depth ($z_H$) suggesting that high oxygen concentrations in the deep waters of lakes were essential for the extent of organic matter mineralization leaving a smaller fraction for anaerobic degradation and thus formation of reduced compounds. Consequently, $F_{red}$ was low in the 310 m deep meso-eutrophic Lake Geneva with high $O_2$ concentrations in the hypolimnion. By contrast, seasonal anoxic conditions enhanced $F_{red}$ in the deep basin of oligotrophic Lake Aegeri. As TOC-MAR and $z_H$ are based on more readily available data, these relationships allow estimating the areal $O_2$ consumption rate by reduced compounds from the sediments where no direct flux measurements are available.

## 1. Introduction

Hypolimnetic oxygen ($O_2$) depletion is a widespread phenomenon in productive lakes and reservoirs. Considerable work has been done to identify parameters responsible for hypolimnetic $O_2$ consumption (Livingstone and Imboden, 1996; Hutchinson, 1938; Cornett and Rigler, 1980), yet the key processes are still debated. Much to the irritation of lake managers, decreasing phosphorus (P) loads to lakes often did not result in a decrease of $O_2$ consumption in the hypolimnion, and $O_2$ consumption even increased in artificially aerated lakes (Müller et al., 2012a). An intuitive explanation for the lack of recovery of $O_2$ consumption is a delay caused by the mineralization of the large amount of organic carbon (OC) deposited in the sediments during hypertrophy, generating reduced species such as $NH_4^+$, $CH_4$, Mn(II), Fe(II) and S(-II). By reacting with $O_2$ and other electron acceptors (directly or via microbial pathways), these reduced species contribute to the hypolimnetic $O_2$ consumption. As direct measurements of reduced substances are rare, several modeling approaches investigated the sediment oxygen demand related to the formation of reduced substances (Di Toro et al., 1990; Soetaert et al., 1996). Further, Matzinger et al. (2010) demonstrated that sediment deposits older than 10 years contributed only ~15% to the areal

hypolimnetic mineralization rate (AHM), thus putting the magnitude and timescale of the "sediment memory effect" into perspective.

Müller et al. (2012a) proposed two key factors to be responsible for the AHM: (i) The diffusion controlled $O_2$ consumption by the mineralization of freshly settled OC at the sediment surface, and (ii) the $O_2$ consumed by the oxidation of reduced

substances diffusing from the sediment ($F_{red}$). The flux of $O_2$ from the bottom water to the sediment surface is a first order process with respect to the concentration of $O_2$ and hence lakes with a large hypolimnion volume can sustain a larger $O_2$ flux and increase the fraction of aerobically mineralized OC. As a consequence, AHM systematically increases with mean hypolimnion depth ($z_H$) of productive lakes. This relationship suggested a constant $O_2$ consumption rate of the sediments, which agreed with the few available estimations from direct sediment porewater measurements of reduced compounds. The

fluxes of $NH_4^+$, $CH_4$, Fe(II), and Mn(II) from eutrophic lakes determined from porewater concentration profiles (summed up and expressed in $O_2$ consuming equivalents) were in a surprisingly narrow range of $0.36 \pm 0.12$ $gO_2$ $m^{-2}$ $d^{-1}$ (Müller et al., 2012a). This can be a substantial fraction of total AHM especially in lakes with a small hypolimnion volume. Matthews and Effler (2006) showed the importance of $F_{red}$ for sediment $O_2$ demand in Onondaga Lake. Further, $F_{red}$ was responsible for up to 42% and 86% of the total AHM in Pfäffikersee and Türlersee (Switzerland), respectively, where $NH_4^+$ and $CH_4$ fluxes

represented up to 90% of $F_{red}$, while Fe(II) and Mn(II) fluxes played only a minor role (Matzinger et al., 2010).

Depending on the sedimentation regime and bottom-water $O_2$ availability, $F_{red}$ is expected to vary spatially. Carignan and Lean (1991) documented that porewater fluxes varied with lake depth and increased with increasing sedimentation rate in a mesotrophic but seasonally anoxic lake. They demonstrated the focusing of labile particulate OC as the cause for the depth dependence. In lakes Baldegg and Sempach, increasing Fe(II) and Mn(II) fluxes with lake depth were attributed to

geochemical focusing (Urban et al., 1997; Schaller et al., 1997). In consequence, extrapolating measurements performed at the deepest sites of lakes to the entire hypolimnion area can significantly overestimate the contribution of reduced sediment compounds to AHM. Hence, the aim of this study is to systematically extend the knowledge of sediment flux measurements of reduced compounds and to identify a common driving factor of their creation. At least three sampling depths were selected in each of the five lakes investigated to gain information on the spatial distribution of fluxes of reduced substances.

The combination of porewater sampling and on-site analysis with two portable capillary electrophoresis systems allowed a high sample throughput and the acquisition of an unprecedented dataset of porewater concentration profiles. Based on observations from 45 cores, this paper assesses the constraints of fluxes of reduced compounds ($CH_4$, $NH_4^+$, Mn(II), and Fe(II)) from the sediments of lakes with a range of trophic histories, discusses their spatial variabilities and the consequences for hypolimnetic $O_2$ consumption.

## 2. Materials and Methods

### 2.1. Study sites

Five lakes of different trophic states and depths were selected for the study (Table 1). Lake Baldegg (66 m depth) is located in an agricultural area dominated by pig farms. After 34 years of artificial aeration and mixing, it is still eutrophic with total
phosphorus (TP) concentrations of ~25 mgP m$^{-3}$. Lake Hallwil is the shallowest of the investigated lakes (48 m) and is presently recovering from its eutrophic past (TP ~12 mgP m$^{-3}$) after 30 years of artificial aeration. Lake Aegeri is oligotrophic (TP ~6 mgP m$^{-3}$) and located in a catchment dominated by pastures and forests. Lake Geneva is the largest lake in Central Europe by volume. It is still meso-eutrophic (TP ~20 mgP m$^{-3}$) and the areal $O_2$ consumption rate is among the highest measured in Swiss lakes (Müller et al., 2012a; Schwefel et al., 2016). Lake Zug (197 m) is eutrophic, permanently
stratified below ~100 m depth (meromictic) and has a TP value of ~30 mgP m$^{-3}$ in the productive epilimnion. In Lake Zug, only one set of cores for porewater analysis, $CH_4$ analysis and bulk sediment parameters was collected from the permanently oxic part (>4 mgO$_2$ L$^{-1}$ throughout the year) at 62 m water depth.

### 2.2 Sediment sampling and porewater analysis

Sediment cores were retrieved with a Uwitec gravity corer equipped with a PVC tube (6.5 cm inner diameter, 60 cm length).
The PVC tube has pre-drilled holes (∅ 2 mm) at 5 mm intervals. The holes were sealed with adhesive tape prior to sampling. Sediment cores were taken along a depth gradient (Table 1). Porewaters were sampled on site immediately after retrieval. 10-50 μL of sediment porewater were retrieved by punctuating the adhesive tape and horizontally inserting a MicroRhizon filter tube (1 mm diameter, 0.20 μm pore size; Rhizosphere Research Products, Wageningen, Netherlands). The sampling resolution was 5 mm for the first 5 cm of sediment, ≤ 1 cm between 5 cm and 10 cm of sediment, ≤ 2 cm between 10 cm and
20 cm of sediment and ≤ 3 cm below 20 cm of sediment. The porewater retrieval time was between 10 to 30 s and samples were immediately analyzed to minimize oxidation. Each porewater sample was analyzed once with two capillary electrophoresis devices each equipped with a capacitively coupled contactless conductivity detector (CE-C$^4$D) (calibrated for anions and cations) directly at the lake shore. Full separation of ions of interest ($NH_4^+$, $Mn(II)$, $Fe(II)$, $SO_4^{2-}$, $NO_3^-$, $NO_2^-$) was achieved within six minutes by applying a voltage of 15 kV and a current of 0.5 μA. The background electrolyte solution and
all calibration standards were freshly prepared before sampling with UltraPure water (Merck) and the corresponding salts. All five-point calibrations were checked against a multi ion standard (Fluka), and standard deviations of all measurements were < 5%. The procedure is described in detail by Torres et al. (2013).

Methane samples were collected from additional sediment cores retrieved on the same day and location. Core liners had holes of 1.2 cm diameter pre-drilled staggered at 1 cm vertical intervals and covered with adhesive tape. Immediately after
retrieval the cores were sampled in 1 cm steps from top to bottom by cutting the tape and inserting a plastic syringe where the tip was cut off. Two cm$^3$ of sediment were transferred into 125 ml serum flasks containing 2 ml of 7 M NaOH and

capped with a septum stopper. Each $CH_4$ sample was analyzed three times in the headspace by gas chromatography (Agilent) using a 1010 Supelco Carboxene column with a standard deviation of 0.1 % to 1.3 %.

Additional sediment cores were extruded and sampled in 0.5 cm to 1 cm sections. Water content was calculated from the weight difference before and after freeze-drying, and the porosity estimated from the density and the respective TOC content (Och et al., 2012). Freeze-dried sediments were ground in an agate mortar and further analyzed for TOC/TN with an ElementAnalyzer Euro EA 3000 (Hekatech). Net sedimentation rates were determined based on the assumption of constant rate of supply with γ-ray measurements of $^{210}$Pb and $^{137}$Cs with a Canberra GeLi borehole detector and/or by varve counting which was possible in all cores except the cores from Lake Hallwil. The net sedimentation rates were further validated by the characteristic $^{137}$Cs peaks of the Chernobyl fallout (1986) and the bomb spike of 1963. As TOC content and net sedimentation rates were not determined from our sediment cores in Lake Geneva, literature data was used to calculate the total organic carbon mass accumulation rate (TOC-MAR, gC m$^{-2}$ yr$^{-1}$). By comparing our coring sites to sites published by Span et al. (1990), Vernet et al. (1983) and Loizeau et al. (2012) we estimated an average net sediment accumulation rate of 1000 g m$^{-2}$ yr$^{-1}$ with a TOC content of 1.1 % resulting in TOC-MAR of 11 gC m$^{-2}$ yr$^{-1}$ for the deep basin. Although TOC content and net sedimentation can vary drastically due to turbidites and the inflow of the Rhone River, we deem this estimate to be representative for the deep undisturbed central basin of Lake Geneva.

## 2.3 Gross sedimentation

Sediment traps were deployed in Lakes Baldegg and Aegeri from March 2013 until the end of November 2014 to determine TOC gross sedimentation rates. In Lake Hallwil sediment traps were deployed from January 2014 to December 2014 (see Suppl. Information Table S1). The sediment trap material was collected biweekly. The traps consisted of two cylindrical PVC tubes with an inner diameter of 9.2 cm and were positioned at 15 m water depth and 1 m above the sediment surface. For the calculation of the gross sedimentation only data from the lower trap was used. The collected material was weighed, freeze-dried and analyzed for TOC and TN with the same methods as the sediment.

## 2.4 Calculation of the flux of reduced compounds

Porewater fluxes were calculated from vertical porewater concentration gradients with a one-dimensional reaction-transport model (Müller et al., 2003) that was adapted from Epping and Helder (1997) and extended from $O_2$ to other parameters. The fluxes ($J$) of reduced compounds ($CH_4$, $NH_4^+$, Fe(II), and Mn(II)), denoted in mmol m$^{-2}$ d$^{-1}$, were multiplied with 32/1000 to be converted into equivalent $O_2$ fluxes (gO$_2$ m$^{-2}$ d$^{-1}$) based on redox stoichiometry and summed up in $F_{red}$ (Eq.1). S(-II) was considered negligible as we detected dissolved Fe(II) in all cores.

$$F_{red} = 2 * J_{CH4} + 2 * J_{NH4} + 0.5 * J_{Mn(II)} + 0.25 * J_{Fe(II)} \tag{1}$$

$F_{red}$ represents the total $O_2$ required per area to oxidize all reduced compounds released by the sediment (Matzinger et al., 2010). As total phosphorus concentration and hypolimnetic $O_2$ consumption rate did not change during the past years, sediment diagenetic processes are assumed to be in quasi steady state. Although seasonally varying deposition rates of OC and varying $O_2$ concentrations may alter Fe(II) and Mn(II) concentration gradients, the fluxes of $NH_4^+$ and $CH_4$ dominating
$F_{red}$ did not change systematically with the seasons.

## 2.5  Estimation of total organic carbon mass accumulation rates

Total organic carbon mass accumulation rate in the lake sediment (TOC-MAR, in gC m$^{-2}$ yr$^{-1}$) at each coring site was calculated from the sedimentation rate (SR, in cm yr$^{-1}$), porosity ($\phi$), dry density ($\rho_{dry}$ in g cm$^{-3}$) and TOC (mg g$^{-1}$) for each 5 mm interval (Och et al., 2012) (Eq. 4). Porosity and dry density are calculated for each sampling interval individually (Eq. 1
and Eq. 3):

$$\phi = V_W \, / \, ( V_w + V_s )  \tag{1}$$

With $V_w$ and $V_s$ being the volumes of water and sediment, while the sediment volume can be calculated from its weight ($W_s$) and the dry density (Eq. 2):

$$V_s \; = \; W_s \, / \, \rho_{dry}  \tag{2}$$

The dry density is estimated according to the empirical relationship between TOC content (in %) and density of geogenic sediments (Och et al., 2012)

$$dry\ density = \, -0.0523 * TOC + 2.65  \tag{3}$$

The TOC-MAR values were then averaged from 2 to 10 cm sediment depth. The first two centimeters were excluded to neglect freshly deposited matter as this material still passes through intense and rapid degradation. The lower TOC-MAR
calculation depth of 10 cm was chosen to remain within the timeframe where steady state conditions can be assumed (Radbourne et al., 2017).

$$TOC - MAR = \, SR * \rho_{dry} \, * (1 - \phi) * 10000 * (TOC/1000)  \tag{4}$$

## 3    Results and Discussion

### 3.1  Porewater concentration profiles and fluxes of reduced compounds

The porewater concentration profiles of the reduced compounds $CH_4$, $NH_4^+$, Fe(II), and Mn(II) measured at different depths in Lake Baldegg, Lake Aegeri, Lake Hallwil and Lake Geneva are presented in Figure 1, and for Lake Zug in Figure S1 (see

Suppl. Information). The highest overall porewater concentrations occurred in Lake Baldegg (Figure 1a) and the lowest in Lake Geneva (Figure 1d) in spite of its high productivity. A distinct pattern of increasing porewater concentrations with increasing sampling depth was apparent in Lake Baldegg and to a lesser extent in Lake Aegeri (Figure 1a, b). Nitrate concentrations in the overlying water of the sediment core were on average 101 µmol L$^{-1}$ in Lake Baldegg, 23 µmol L$^{-1}$ in

Lake Aegeri, 61 µmol L$^{-1}$ in Lake Hallwil, 32 µmol L$^{-1}$ in Lake Geneva and 20 µmol L$^{-1}$ in Lake Zug and steeply declined to zero within the first cm of the sediment.

The trend of increasing ion concentrations with lake depth observed in the porewater was also reflected in the areal fluxes of the reduced compounds from the sediment to the lake bottom waters (Table 2). The fluxes were positive from the sediment to the bottom water in all lakes at all depths. Fluxes measured at the same locations on up to five different dates between

March and October varied by 34% in Lake Baldegg and by 84% in Lake Aegeri for $CH_4$, and by 80% in Lake Baldegg and by 65% in Lake Aegeri for $NH_4^+$. The fluxes of Mn(II) (66% and 72%) and Fe(II) (46% and 88%) also showed a high variability. This temporal variation is likely due to local heterogeneity of the sediment and seasonal variations of both the supply of OC, e.g, algae blooms, and the $O_2$ concentration at the sediment-water interface. However, the relative importance of these driving factors could not be determined, and no clear seasonal pattern was detected.

A summary of all fluxes and $F_{red}$ is given in Table 2. In Lake Baldegg the highest fluxes of $CH_4$, $NH_4^+$, and Fe(II) of all lakes, and a clear increase with sampling depth were observed. Fluxes at the deepest site agreed well with earlier measurements from dialysis samplers (Urban et al., 1997). In the oligotrophic Lake Aegeri fluxes of $CH_4$ and $NH_4^+$ were small at the shallow sites and similar to those observed in other oligotrophic lakes (Frenzel et al., 1990; Carignan et al., 1994), while at the deepest location considerably higher fluxes were measured for $CH_4$. Fluxes in Lake Hallwil did not show

an overall increase with lake depth. In Lake Geneva, the smallest fluxes of $CH_4$, $NH_4^+$, and Fe(II) were observed in spite of its high productivity, without systematic variations with lake depth. $NH_4^+$ and $CH_4$ contributed 85% to 98% to the $O_2$ consuming capacity while Fe(II) and Mn(II) played only a minor role. Müller et al. (2012a) estimated $F_{red}$ for a range of eutrophic lakes to be 0.36±0.12 gO$_2$ m$^{-2}$ d$^{-1}$, based on a relationship between hypolimnetic $O_2$ consumption rates and mean hypolimnion depths. $F_{red}$ values observed at 24 m (0.28 gO$_2$ m$^{-2}$ d$^{-1}$) and 40 m (0.34 gO$_2$ m$^{-2}$ d$^{-1}$) depth in the eutrophic Lake

Baldegg (see Table 2) matched the modeled value. $F_{red}$ at the deepest site of Lake Baldegg (0.49 gO$_2$ m$^{-2}$ d$^{-1}$) agreed with a previous observation of 0.55 gO$_2$ m$^{-2}$ d$^{-1}$. Likewise in Lake Hallwil, $F_{red}$ varied between 0.18 gO$_2$ m$^{-2}$ d$^{-1}$ and 0.25 gO$_2$ m$^{-2}$ d$^{-1}$, matching earlier measurements of 0.28 gO$_2$ m$^{-2}$ d$^{-1}$ (Müller et al., 2012a). In Lake Aegeri, $F_{red}$ was clearly higher at the deepest sampling site than at the shallower sites. At 34 m and 49 m water depth, $F_{red}$ was 0.07 gO$_2$ m$^{-2}$ d$^{-1}$, and thus typical for a deep oligotrophic lake. At 79 m, a markedly higher $F_{red}$ of 0.26 gO$_2$ m$^{-2}$ d$^{-1}$ was observed. In eutrophic Lake Geneva,

$F_{red}$ varied between 0.02 and 0.09 gO$_2$ m$^{-2}$ d$^{-1}$, which is surprisingly low for a productive lake. In summary, we did not observe a direct relationship between the trophic state of the lake and the fluxes of reduced substances.

Determination of sedimentation rates from core dating revealed increasing sediment deposition with increasing lake depth in Lake Baldegg and at the deepest site of Lake Aegeri. The sediment TOC content was around 3.4 % and varied only little between the different coring sites and lakes (see Table 2), except for Lake Geneva with an estimated 1.1% TOC. Consequently, mass accumulation rates varied substantially and increased with depth in Lake Baldegg and to a lesser extent in Lake Aegeri. We attribute this observation to sediment focusing, which has also been documented by Urban et al. (1997) for Lake Baldegg. Sediment focusing transports fine, freshly settled organic rich material from the shallower to the deeper parts of a lake and consequently increases TOC-MAR with lake depth (Lehman, 1975). Carignan and Lean (1991) documented that porewater fluxes increased with lake depth caused by the focusing of labile particulate OC into the deeper part of the lake. A study at oligotrophic Little Rock Lake also showed elevated $NH_4^+$ concentrations at the deepest site due to a greater supply of fine-grained organic particles caused by sediment focusing (Sherman et al., 1994). In eutrophic Lake Zug, Maerki et al. (2009) found that $NH_4^+$ fluxes increased proportionally with the sediment contents of TOC and total nitrogen (TN), indicating a link between fluxes of reduced substances and TOC-MAR. In Lakes Baldegg and Hallwil, geochemical focusing, caused by recurring redox-sensitive dissolution and precipitation of Mn and Fe phases, is an additional process that increases Fe(II) and Mn(II) concentrations with lake depth (Urban et al., 1997; Schaller and Wehrli, 1996).

Sediment focusing increased TOC-MAR by ~104% in the deepest part of Lake Baldegg and by ~43% in Lake Aegeri. Since $F_{red}$ depends on the sedimentation regime and bottom-water $O_2$ availability, this explains the spatial variability of $F_{red}$ in these two lakes. No sediment focusing was observed in Lake Hallwil which is in agreement with a previous study by Bloesch and Uehlinger (1986). In consequence, extrapolating measurements performed at the deepest sites of lakes to the entire lake area can significantly overestimate the average contribution of reduced sediment compounds to AHM in case sediment focusing is active.

## 3.2 $F_{red}$ controlled by sediment TOC mass accumulation rate

All lakes investigated have a predominance of autochthonous OC input, implied by similar C/N ratios (7.0 – 9.9), a proxy for the origin of OC (see Table 2), and a permanently oxic hypolimnion (see Suppl. Information Fig S2). As a consequence, the burial efficiency of OC, defined as the ratio between TOC-MAR and OC gross sedimentation rate (deposition rate of OC onto the sediment surface) by Sobek et al. (2009) should be rather similar in these lakes. Based on gross TOC sedimentation data from sediment traps (see Suppl. Information Table S1) and TOC-MAR values (Table 2), burial efficiencies at the deepest sampling locations were calculated from sediment trap data of Lakes Baldegg (50 %), Hallwil (41 %) and Sempach (46 %). All values were close to the average value of 48% determined from 27 sediment cores from 11 lakes by Sobek et al. (2009). Consequently, a similar proportion of gross OC sedimentation is buried and contributes to the formation of $F_{red}$ and TOC-MAR in all studied lakes. An exception is Lake Aegeri, with a surprisingly high burial efficiency of OC of 77 % calculated for the deepest site. However, this is caused by the exceptional bathymetry. The deepest site is located in a small trough with surrounding steep slopes predestined for sediment slides and remobilization of settled particles. The locally high

ratio of sediment area to water volume presumably leads to the annual development of an anoxic deep water layer which increases OC burial, but is not representative for the whole lake.

As primary production and hypolimnetic $O_2$ concentrations did not change considerably during the last decade, the burial efficiency and thus TOC-MAR and $F_{red}$ generation likely remained unchanged. Furthermore, porewater profiles do not
capture the effects of rapid initial mineralization occurring within the top few millimeters of the sediment, but mirror the slower processes of anaerobic degradation of buried OC and $F_{red}$. Hence, in Figure 2 we related $F_{red}$ to the corresponding TOC-MAR at each sampling location. Additional datasets from earlier measurements in various lakes were added to complement the relationship.

Figure 2 reveals two characteristic facts concerning the release of reduced compounds from lake sediments: (i) A distinct
increase of $F_{red}$ was observed when TOC-MAR exceeded ~10 gC m$^{-2}$ yr$^{-1}$, (ii) $F_{red}$ increased proportionately with TOC-MAR between 10 and 45 gC m$^{-2}$ yr$^{-1}$ up to ~0.50 gO$_2$ m$^{-2}$ d$^{-1}$ in all seasonally mixed lakes investigated. The highest $F_{red}$ value was measured at the deepest point of Lake Baldegg with 0.49 gO$_2$ m$^{-2}$ d$^{-1}$ at a TOC-MAR of 45 gC m$^{-2}$ yr$^{-1}$, despite much higher TOC-MAR in Rotsee. The mineralization of sediment OC appeared to be the main driver of $F_{red}$ independent of the cause of TOC accumulation. The areal accumulation of TOC per time is controlled by gross sedimentation (which is related to
primary production), $O_2$ concentration in the bottom water, biological factors like grazing and bioturbation, and physical parameters such as sediment focusing (Sobek et al., 2009). At low TOC-MAR, the total flux of reduced substances was very low (e.g Lakes Baikal, Erie and Superior, Figure 2), as only little carbon remained for anaerobic degradation, and the reduced substances diffusing up from deeper sediment strata were quantitatively oxidized in the upper sediment layers.

### 3.3 Factors limiting $F_{red}$ and TOC-MAR

Given that all investigated lakes are seasonally mixed and have a permanently oxic hypolimnion, the likely driving factors for the positive relationship between $F_{red}$ and TOC-MAR are (i) hypolimnetic $O_2$ concentrations, and (ii) the quality and quantity of OC. An influence of temperature and benthic production can be ruled out as all sampling stations were located in the cold hypolimnia well below the thermocline. Generally, high $O_2$ concentrations lead to a high fraction of aerobic OC mineralization and hence decrease of TOC-MAR as OC is decomposed by oxygenases and other reactive oxygen species
(Maerki et al., 2006; Sobek et al., 2009; Stumm and Morgan, 1996). Furthermore, elevated hypolimnetic $O_2$ concentrations increase the oxidation of reduced compounds near or within the top sediment layer and thus increase the regeneration of alternative electron acceptors such as $NO_3^-$ and $SO_4^{2-}$ (Urban et al., 1997). In contrast, low $O_2$ concentrations or even temporally anoxic conditions increase OC burial and thus TOC-MAR and prompt the production of reduced compounds (Sobek et al., 2009).

Low TOC-MAR occurred in lakes with low primary production and low allochthonous input. In the oligotrophic Lakes Superior and Baikal, TOC-MAR was 4 and 7 gC m$^{-2}$ yr$^{-1}$ (see Suppl. Information Table S2), respectively, and the resulting

$F_{red}$ from the sediment was close to zero (Och et al., 2012; Klump et al., 1989; Remsen et al., 1989; Richardson and Nealson, 1989). In addition to low gross sedimentation of OC, the high sediment $O_2$ penetration depth (of around 1-3 cm) causes a long exposure time to oxic conditions and thus oxic mineralization of a large fraction of the deposit (Maerki et al., 2006; Martin et al., 1993; Li et al., 2012). In consequence, the TOC buried in lakes like Superior and Baikal is already highly mineralized and therefore does not generate significant amounts of reduced substances. However, low TOC-MAR were also observed in Lake Geneva. Although Lake Geneva is highly productive, its hypolimnetic $O_2$ concentration remained high throughout the year (see Suppl. Information Fig. S2) (Schwefel et al., 2016). Randlett et al. (2015) concluded that ~75% of the OC in Lake Geneva was mineralized aerobically at the sediment surface. Measurements performed by Schwefel et al. (2017) further confirmed that >96% of the OC in Lake Geneva is mineralized aerobically within the water column or at the sediment surface. As the buried OC only generated low $F_{red}$ of 0.03 to 0.09 $gO_2$ $m^{-2}$ $d^{-1}$ (see Table 2), we concluded that the material was already recalcitrant. In contrast to the other lakes investigated, in Lake Geneva high porewater concentration and sediment penetration of $SO_4^{2-}$ enhanced the degradation of OC and actively diminished $F_{red}$ by oxidation of $CH_4$ and formation of Fe-sulfides. Norði et al. (2013) showed the efficiency of anaerobic $CH_4$ oxidation by $SO_4^{2-}$ and a reactive Fe(III) pool which in turn reduced the flux of $CH_4$ out of the sediment. Sediment core measurements at 210 m and 240 m but in proximity to the Rhone River delta by Randlett et al. (2015) showed that at even higher TOC-MARs of 20 to 30 gC $m^{-2}$ $yr^{-1}$ (two to three times the rate estimated for the open lake), $F_{red}$ values remained at similarly low values of 0.04 to 0.05 $gO_2$ $m^{-2}$ $d^{-1}$. This case highlights the importance of OC quality, as refractory OC can be sequestered and offset TOC-MAR without a noticeable increase of $F_{red}$. Consequently, lakes with a higher input of land derived organic material should show a higher offset in TOC-MAR values as more recalcitrant OC is buried without a direct effect on $F_{red}$. While TOC-MAR values were similar at all three stations in Lake Aegeri, $F_{red}$ peaked at the deepest point. In addition to a likely sediment focusing, a small anoxic bottom layer developed at the end of summer stratification at the deepest location (see Suppl. Information Fig. S2) due to the steep topography. This condition diminishes oxic mineralization of settled OC and thus supports the formation of higher $F_{red}$.

Under the assumptions that primary production even under ideal circumstances can generate only a limited amount of OC of around ~400 to 500 gC $m^{-2}$ $yr^{-1}$ (Wetzel, 2001) and that allochthonous, soil and land-plant derived OC is comparatively less accessible for mineralization, $F_{red}$ is expected to converge at an upper bound even at high total TOC-MAR. As shallower lakes with a high primary production tend to become anoxic during the stratification period and thereby start accumulating reduced substances in the deepest part of the hypolimnion, $F_{red}$ should not further increase as the concentration gradients between sediment and water would flatten as the oxic-anoxic interface moves from the sediment into the bottom water. The high $F_{red}$ values encountered in Lake Baldegg (~0.49 $gO_2$ $m^{-2}$ $d^{-1}$) supposedly represent an upper boundary of $F_{red}$, as it is an example of a highly eutrophic lake with an additional supply of OC to the deepest part by sediment focusing while only retaining an oxic hypolimnion due to artificial aeration. At the highest TOC-MAR value of ~170 gC $m^{-2}$ $yr^{-1}$ measured in the seasonally anoxic Rotsee (RO), $F_{red}$ remained at 0.46 $gO_2$ $m^{-2}$ $d^{-1}$ (measured after lake mixing in the oxic hypolimnion),

showing no further increase of $F_{red}$, while measurements performed at the end of summer stagnation and an anoxic hypolimnion revealed a $F_{red}$ value of 0.26 $gO_2$ $m^{-2}$ $d^{-1}$. However, whether $F_{red}$ consistently levels off at high TOC-MAR remains to be verified by measurements in additional eutrophic lakes with an oxic hypolimnion and high TOC-MAR.

### 3.4 Relationship between $F_{red}$ and mean hypolimnion depth

The increasing fraction of aerobically mineralized OC with increasing $O_2$ availability in eutrophic lakes is further supported by a systematic decrease of $F_{red}$ with increasing mean hypolimnion depth ($z_H$) in productive lakes, shown in Figure 3. During the stratified period, the hypolimnetic $O_2$ reservoir in eutrophic lakes with a small $z_H$ is quickly exhausted, enforcing a higher OC burial rate, increased anaerobic mineralization, and thus the formation of reduced substances e.g. in Rotsee, Türlersee and Pfäffikersee. In these lakes $F_{red}$ becomes the dominant fraction of AHM with values of ~0.40 $gO_2$ $m^{-2}$ $d^{-1}$. In the deep
Lake Geneva, the hypolimnetic $O_2$ inventory increases with $z_H$, and the lakes' resilience to $O_2$ depletion rises. Consequently, more $O_2$ is available for aerobic remineralization of OC, and hence less or more degraded OC is buried. Hence $F_{red}$ diminishes with increasing $z_H$. Coherently, very deep eutrophic lakes such as Lake Geneva are well protected from anoxia. Lake Baldegg deviates from this general correlation in Figure 3 due to the high sediment focusing, which caused the sedimentation rate to increase by a factor of 1.9 compared to the shallower sites. Likewise, sediment focusing might increase
$F_{red}$ in other lakes. Yet it is unclear to what extent sediment focusing increases $F_{red}$ and TOC-MAR, for example, in Lake Sempach (Urban et al., 1997). These findings complement and extend the observation presented in Müller et al. (2012a) that AHM of fully productive lakes increased linearly with their mean hypolimnion depth if $z_H < ~25$ m, with a similar contribution of reduced compounds from the sediments of all lakes. A closer look on the fluxes of reduced compounds produced by the deposited organic matter in the sediment, however, revealed that they as well depend on the concentration
of $O_2$ that the material was exposed to.

## 4    Conclusion

We demonstrate that the areal oxygen consumption in lakes caused by reduced compounds diffusing from the sediment, $F_{red}$, is strongly related to the local mass accumulation rate of OC (Figure 2). In fully productive eutrophic lakes, the flux of reduced compounds, $F_{red}$, declines with increasing mean hypolimnion depth ($z_H$) due to the higher $O_2$ bottom water
concentration and thus increasing exposition time of settled OC to $O_2$ (Figure 3). Hence, in these lakes, $z_H$ can serve as a proxy for $F_{red}$. These observations indicate that $F_{red}$ from the sediment is constrained mainly by the deposition rate and quality of OC, $O_2$ availability to the sediment surface and lake bathymetry (i.e. the occurrence of sediment focusing). The sediment $O_2$ demand, a major sink for $O_2$ in the hypolimnion, can now be estimated for a broad range of lakes with a permanently oxic hypolimnion without elaborate $O_2$ measurements at the sediment-water interface based on the relationships
between $F_{red}$ and TOC-MAR and between $F_{red}$ and $z_H$, which are more commonly available than porewater measurements.

## 5    Data Availability Statement

The data will be made available over Figshare.com

## 6    Appendices

5    Table S1 summarizes parameters used for OC burial efficiency rate calculations e.g. TOC-MAR values and benthic OC gross sedimentation. Table S2 sums up information about $F_{red}$ and TOC-MAR values of lakes taken or calculated from literature. Figure S1 shows the porewater concentrations of $CH_4$, $NH_4^+$, Fe(II) and Mn(II) in the sediment of Lake Zug at 62 m water depth, Figure S2 depicts the $O_2$ concentrations throughout one year in five different lakes at the core sampling depths.

## 10    Author contribution statement

MS, AW, BW and BM designed the study. TS conducted porewater measurements and analyzed the sediment cores and wrote the manuscript. RS performed $O_2$ microprofiles. All authors contributed to the writing of the manuscript.

## Competing interests

The authors declare that they have no conflict of interest.

## 15    Acknowledgements

We thank Robert Lovas (Environment and Energy, Canton of Lucerne) for providing data of Lakes Sempach and Baldegg. Arno Stöckli (Dept. for the Environment, Canton of Aargau) is acknowledged for providing monitoring data of Lake Hallwil. Further, we acknowledge Peter Keller (Dept. for the Environment, Canton of Zug) for providing monitoring data of Lake Aegeri. $O_2$ data for Lake Geneva was provided by the Commission International pour la Protection des Eaux du Léman 20    (CIPEL) and the Information System of the SOERE OLA (http://si-ola.inra.fr), INRA Thonon-les-Bains. We thank Patrick Kathriner for the great help in the laboratory and on field campaigns. The valuable comments and suggestion by the reviewers greatly helped to clarify and improve this paper. This work was financially supported by SNF Grant 200021_146234.

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

25

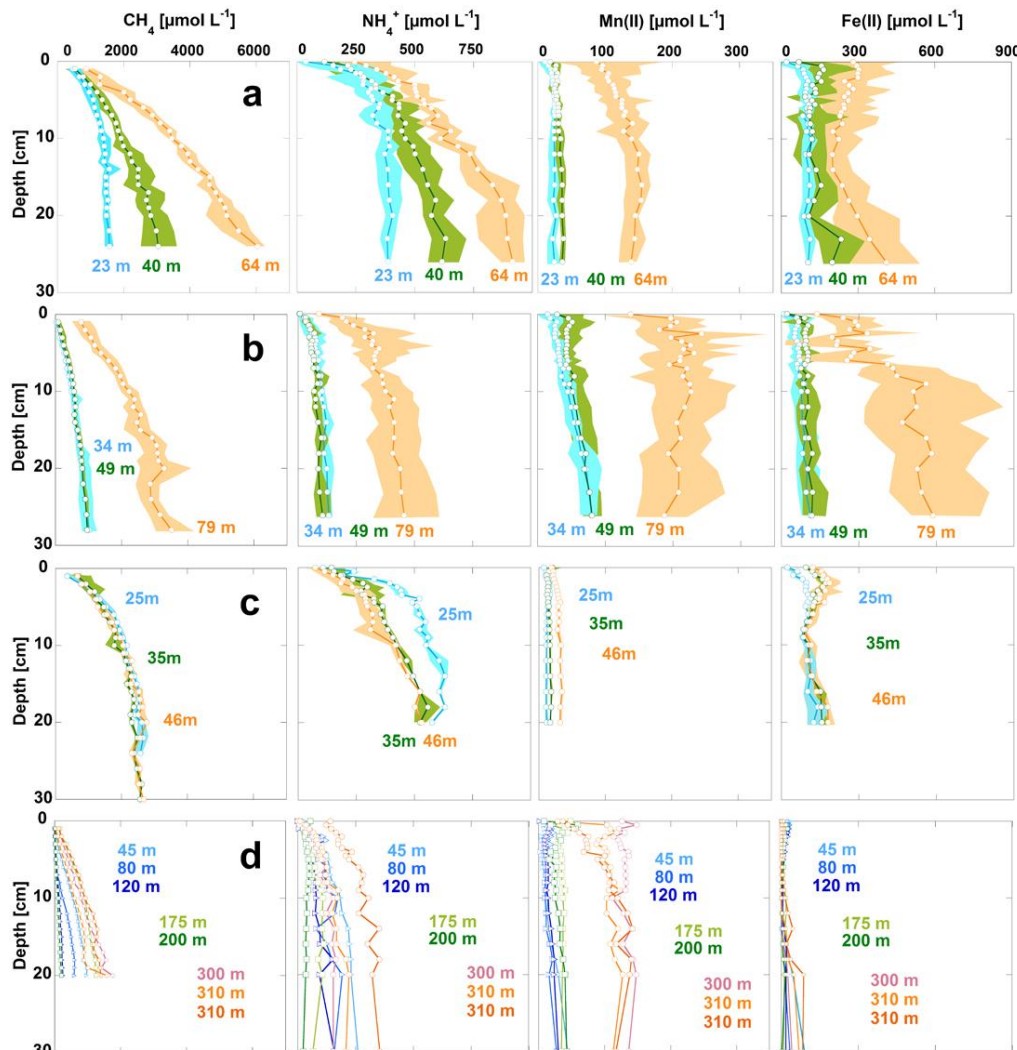

**Figure 1.** Porewater concentration profiles of $NH_4^+$, $CH_4$, Mn(II), and Fe(II) from a) Lake Baldegg, b) Lake Aegeri, c) Lake Hallwil, and d) Lake Geneva. Bold lines are averaged values of up to five measurements while the areas of corresponding colors show the range of minimal and maximal values. Coring sites of Lake Geneva (d) were sampled only once.

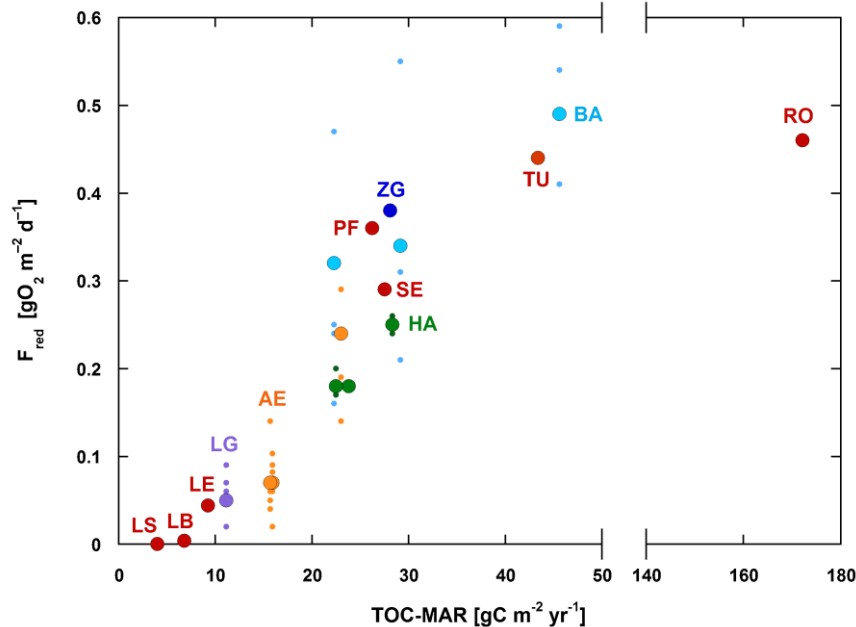

**Figure 2.** $F_{red}$ of 11 different lakes plotted against the total organic carbon mass accumulation rate (TOC-MAR). Values for Lakes Baldegg (BA), Hallwil (HA) and Aegeri (AE) were averaged from up to five measurements (big circles), those for Lake Zug (ZG) were calculated from a single core at 62 m water depth. Small circles show each individual $F_{red}$ result at the respective sampling location. The variations of

5  $F_{red}$ in Lake Geneva show only the variations due to sampling depths as all cores were collected in summer. Red marks were calculated from single core literature data (Lake Baikal (LB), Och et al. (2012); Lake Sempach (SE), Müller et al. (2012b); Rotsee (RO), Naeher et al. (2012); Pfäffikersee (PF, unpublished); and Türlersee (TU, unpublished)). TOC-MAR from Lake Geneva (LG) are based on sedimentation rate estimates from the literature (Vernet et al., 1983; Loizeau et al., 2012; Span et al., 1990). TOC-MAR and $F_{red}$ values from Lake Erie (LE) were extracted from Matisoff et al. (1977), Adams et al. (1982) and Smith and Matisoff (2008). Values for Lake Superior (LS) were

10  compiled from Klump et al. (1989), Remsen et al. (1989), Richardson and Nealson (1989), Heinen and McManus (2004) and Li et al. (2012).

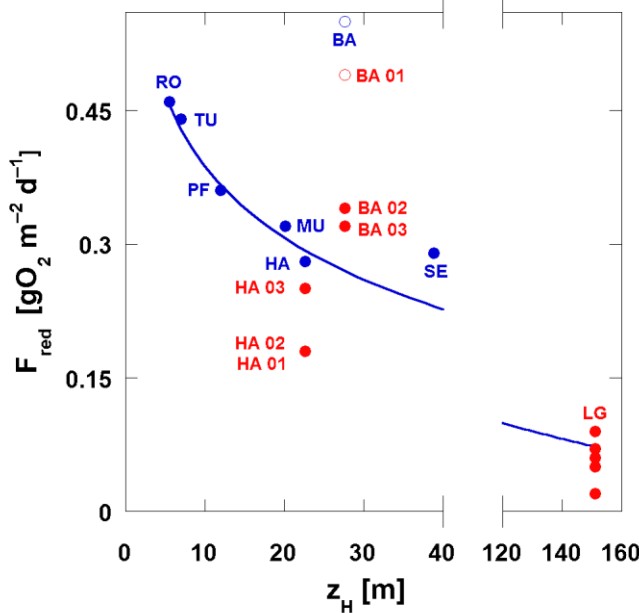

**Figure 3.** Average fluxes of reduced compounds from the sediments of eight meso- to eutrophic lakes decrease systematically with mean hypolimnion depth ($z_H$), except for sediments sampled at the deepest sites of lakes with pronounced sediment focusing (BA, BA 01, open circles). $F_{red}$ values of Lake Baldegg (BA01 to 03) show a significant increase with sampling depth due to strong sediment focusing. Blue circles indicate data for Rotsee (RO), Türlersee (TU), Pfäffikersee (PF), Lake Murten (MU), and Lake Sempach (SE), taken from Müller et al. (2012a) while red circles show data of the current study. Meromictic Lake Zug is not shown. The blue line is used for eye guidance.

**Table 1.** Lake characteristics, sampling depths and dates. The mean hypolimnion depth ($z_H$) is defined as the hypolimnetic volume divided by the hypolimnetic area below 15 m water depth.

| Lake | Tropic status | Hypolimnion surface area, (km$^2$) | Hypolimnion volume ($10^6$ m$^3$) | $z_H$ (m) | Max. depth (m) | Sampling depths (m) | Sampling time |
|------|---------------|-----------------------------------|-----------------------------------|-----------|----------------|---------------------|---------------|
| Lake Baldegg | eutrophic | 4.53 | 125 | 27.6 | 66 | 23, 40, 64 | March 2013, May 2013, August 2013, October 2013, March 2014, |
| Lake Aegeri | oligotrophic | 6.64 | 283 | 42.6 | 81 | 34, 49,79 | March 2013, May 2013, August 2013, October 2013, March 2014, |
| Lake Hallwil | Mesotrophic recovering | 8.58 | 194 | 22.6 | 48 | 25, 35, 46 | April 2014, August 2014 |
| Lake Zug | eutrophic - meromictic | 34.5 | 2660 | 77.1 | 197 | 62 | Mai 2016 |
| Lake Geneva | Meso-eutrophic | 534 | 80800 | 151 | 310 | 45**, 80*, 120**, 175*, 200*, 300*, 310* and ** | *July 2014, ** July 2015 |

5     **Table 2.** Results from sediment and porewater analyses. Porewater fluxes are averaged over all flux measurements of each individual species. Standard deviations ($\pm$) are based on all measurements. TOC was averaged from 2 to 10 cm sediment depth. TOC and net sedimentation from Lake Geneva were not determined (n.d). Lake Baldegg (BA), Lake Aegeri (AE), Lake Hallwil (HA), Lake Zug (ZG) and Lake Geneva (LG).

| Core | Depth (m) | SR (mm yr$^{-1}$) | TOC (%) | C/N ratio | $J_{NH4}$ | $J_{CH4}$ | $J_{Fe(II)}$ | $J_{Mn(II)}$ | $F_{red}$ (gO$_2$ m$^{-2}$ d$^{-1}$) | TOC-MAR (gC m$^{-2}$ yr$^{-1}$) | No. Cores |
|------|-----------|-------------------|---------|-----------|-----------|-----------|--------------|--------------|---------|----------|-----------|
| | | | | | | (mmol m$^{-2}$ d$^{-1}$) | | | | | |
| BA 03 | 23 | 1.75 | 2.70 | 7.5 | 2.04±1.62 | 2.16±0.74 | 0.34±0.14 | 0.05±0.03 | 0.28±0.11 | 22.3 | 5 |
| BA 02 | 40 | 2.63 | 2.63 | 7.0 | 2.19±1.34 | 2.79±0.84 | 0.40±0.08 | 0.07±0.03 | 0.34±0.14 | 29.1 | 5 |
| BA 01 | 64 | 3.32 | 3.42 | 7.5 | 2.84±1.23 | 4.24±0.57 | 1.00±0.39 | 0.22±0.11 | 0.49±0.09 | 45.6 | 5 |
| AE 03 | 34 | 1.43 | 3.49 | 8.1 | 0.45±0.25 | 0.62±0.42 | 0.33±0.29 | 0.08±0.04 | 0.07±0.04 | 16.0 | 5 |
| AE 02 | 49 | 1.37 | 3.47 | 7.6 | 0.50±0.21 | 0.49±0.41 | 0.34±0.21 | 0.18±0.13 | 0.07±0.03 | 16.3 | 5 |
| AE 01 | 79 | 1.91 | 3.43 | 7.9 | 1.44±0.94 | 2.06±0.99 | 0.69±0.32 | 0.41±0.17 | 0.26±0.08 | 22.8 | 5 |
| HA 03 | 25 | 2.00 | 3.46 | 9.9 | 1.58 | 2.26 | 0.28 | 0.03 | 0.25 | 28.3 | 2 |
| HA 02 | 35 | 1.93 | 3.42 | 9.5 | 1.18 | 1.64 | 0.40 | 0.04 | 0.18 | 23.8 | 2 |
| HA 01 | 46 | 1.95 | 3.41 | 9.9 | 0.98 | 1.74 | 0.53 | 0.05 | 0.18 | 22.5 | 2 |
| ZG | 62 | 2.80 | 3.99 | 7.6 | 2.80 | 3.11 | 0.24 | 0.03 | 0.38 | 28.1 | 1 |
| LG 08 | 45 | n.d | n.d | n.d | 0.26 | 0.45 | 0.18 | 0.01 | 0.05 | n.d | 1 |
| LG 07 | 80 | n.d | n.d | n.d | 0.42 | 0.41 | 0.14 | 0.03 | 0.05 | n.d | 1 |
| LG 06 | 120 | n.d | n.d | n.d | 0.16 | 0.11 | 0.16 | 0.05 | 0.02 | n.d | 1 |
| LG 05 | 175 | n.d | n.d | n.d | 0.15 | 0.87 | 0.03 | 0.05 | 0.07 | n.d | 1 |
| LG 04 | 200 | n.d | n.d | n.d | 0.13 | 0.08 | 0.11 | 0.04 | 0.02 | n.d | 1 |
| LG 03 | 300 | n.d | n.d | n.d | 0.21 | 0.61 | 0.00 | 0.15 | 0.05 | n.d | 1 |
| LG 02 | 310 | n.d | n.d | n.d | 0.30 | 0.54 | 0.04 | 0.21 | 0.06 | n.d | 1 |
| LG 01 | 310 | n.d | n.d | n.d | 0.52 | 0.79 | 0.12 | 0.16 | 0.09 | n.d | 1 |