# Peer review of "Organic carbon mass accumulation rate regulates the flux of reduced substances from the sediments of deep lakes"

_Biogeosciences, 2017_

## Referee Comment (RC1) · Anonymous Referee #1 · 30 Mar 2017

Steinsberger et al. investigate the flux of reduced substances (Fred) from sediments in five deep lakes of different trophic states. They found no indication that the trophic state of the lakes controls Fred. However, organic carbon mass accumulation rates together with the mean hypolimnion depth of the lakes relate to Fred and can potentially be used to estimate the influence of Fred to O2 consumption in eutrophic deep lakes. The authors collected a big dataset from five deep lakes and calculate/estimate some factors that are relevant for assessing the hypolimnetic O2 consumption and its driving factors.

The overall topic of the paper falls into the scope of Biogeosciences and it presents some novel and a solid dataset to assess the fluxes of reduced compounds from sediments and its role for hypolimnetic oxygen consumption. The overall presentation of the methods and results/discussion could be improved. The whole manuscript seems lengthy and quite descriptive at many places. I will elaborate on this in more specific comments below. I would recommend a publication but only after a rewriting of the aims/hypothesis and methods and a restructuring of the discussion to strengthen the main messages.

Major comments:

1) The aims of the paper could be reformulated to increase the curiosity of the reader. As it is now, the aims are: extend a dataset/assess constraints/discuss spatial variabilities and consequences... This is also reflected in the results/discussion section that is often hard to follow and difficult to say what the authors want to say/conclude here. The first two paragraphs in the "Results and Discussion" (page 5) are only data descriptions without any interpretations jumping from one lake to the other. I got easily lost in the details and did not get the major results and their interpretation, something that I would expect at the beginning of this section. I would suggest to reformulate the aims and maybe try to formulate a hypothesis (or hypotheses) or expectations from the data and analyses. With those newly formulated hypotheses the "Results and Discussion" section should be rewritten/-structured, focusing on the new hypotheses.

2) The "Materials and Methods" description has missing information:

How many cores were taken per day and depth? I am confused because the authors talk about a "set of cores" collected in Lake Zug (p. 3 line 2). Was there any replication or does this refer to the three cores taken for all analyses including reduced substances via capillary electrophoresis, methane and water/TOC content? In the introduction, the authors talk about 50 cores that they took (p. 2 line 21). When I count one core per date and depth for the five lakes (Table 1), I get to 57 cores, which means no replication. How reliable are those data without replication? And what happened to the 7 cores that do not match with the number stated and my calculations?

**BGD**

It would also be nice to read somewhere how many times the reduced substances via capillary electrophoresis, methane and water/TOC content in each core were measured and to what depth. The distances of the holes are mentioned I could not figure out how deep the sediment was, only from looking at figure 1. From Figure 1, I can also see that it has different numbers (by counting dots) and sometimes different depths and that the distances between points change. This is not mentioned at all in the method description. I would like to see a photograph of the cores with the holes, maybe added to the supplement. That would make it much easier to picture such cores.

A short description about the literature search in the main text would be helpful. How did the authors search for those data and what did they extract and did they all use similar methods?

3) Assessment of uncertainty of data: The authors provide only limited information on the range of their data. I already asked the question if the authors replicate the sampling at one point on one sampling day and if not how reliable the data are. In figure 1, there are ranges of the data and you can see that especially at the deepest points, there are wide ranges. But in table 2, there is only one value. Did the authors calculate averages for the sampling times? Or are these data from only one sampling time? It is hard to assess the variability of the data at each sampling point without any knowledge of variation or uncertainty analysis. The authors do not test their results!

4) I miss some references throughout the text:

p. 2 line 5: "This relationship suggested a constant fraction of O2 consumption from the sediments, which agreed with the few available estimations from direct of sediment porewater measurements of reduced compounds (ref.)." p. 3 line 17-21: a reference for the headspace technique? p. 4 line 26: "The lower TOC-MAR calculation depth of 10 cm was chosen to remain within the timeframe were steady state conditions can be assumed (ref.)." p. 7 line 17-19: "The areal accumulation of TOC per time is controlled by gross sedimentation (which is related to primary production), O2 concentration in

the lake bottom water, biological factors like grazing and bioturbation, and physical parameters such as sediment focusing (ref.)." Or is this results taken from the author's own data?

5) Why did the authors install the sediment traps at 15 m water depth? All sampling points of the cores are at deeper points and the sedimentation can change with deeper waters, especially because 15 m water depth is above the hypolimnion in most lakes. Does this influence the data and conclusions? Does it play a role and if yes, how? Please also consider discussing this in the main text.

Minor points: - P. 2 line 4: . . .from direct sediment porewater. . . Delete "of"! - P. 5 line 4: in the four lakes . . . No capital letter! - P. 8 line 3: do you need to say "from the sediments were virtually zero"? Do the authors refer to both lakes that they mention before or only one here? - P. 10 line 3: "more commonly available than"

---

## Referee Comment (RC2) · Anonymous Referee #2 · 19 Apr 2017

The paper by Steinsberger et al presents interresting results on the role of dissolved fluxes from sediment in the oxygen consumption in lakes with oxic hypolimnion. It fits perfectly one of the scopes of the journal, linking mainly chemical and physical aspects of the cycle of chemical substances, organic matter, and sedimentation rates. The paper presents new data from fives Swiss lakes with different trophic status. Results, interpretation and conclusion seem coherent, however my main comment concerns i) the lack of clarity in the presentation of the results, and their use in figures. For instance 8 cores were collected in Lake Geneva with corresponding Fred, but only one point plotted (and discussed?) on figure 2 and 3 (average value, deepest point?). I have the general feeling that a large set of data has been produced, but partly discussed; and ii) how the variability in the observed fluxes is taken into consideration in the final assessement. On page 5 line 14 and following, the authors correctly indicate that fluxes, at the same location, show variations due to local sediment heterogeneity and/or seasonal effect. Depending on the substances, values varies between 23% to 67%. However, only one value per lake /depth is given in table 2, without any uncertainty, either from the measurements themselves (including uncertainty in sediment accumulation rates) or from the replicates. Then how the values in table 2 are computed (simple average, time weighted)? What could have these uncertainties on the interpretation and conclusion? From a quick evaluation it seems that the main trends are still significant, but this should be discusssed in the manuscript to improve the strength of the conclusion. More detailed comments: Page 2 line 31. From the classical reference (Wetzel 2001), Lake Geneva is meso-eutrophe (10-30 mg/m3) based on phosphorus content (20 mg/m3), but also on chlorophyll (.

P4 line 23. I don't understand why the sedimentation rate (SR) is calculated based on a depth scale, and then at each layer a TOC-MAR (mass accumulation rate) is calculated, including porosity and dry density. This way is correct if the porosity is relatively constant downcore. But in general in recent sediment porosity vary strongly with depth, and this variation should be taken into account before the computation of the sediment rate. For instance a SR of 2 mm/y correspond to 0.05g cm/2/y with 90%porosity, but 0.1 g/cm2/y with 80% porosity.

P4 line 24. It is not clearly explain here (but discussed later) why the surface sediments are excluded from the computation.

P6 line 28. Not clear here the difference between TOC-MAR and OC (or TOC?) gross sedimentation rate.

P9 line 22. Not clear what is meant by "accessibility of hypolimnetic O2 to the sediment surface".

Table 1. Units of Hypolimnetic volume is (Mm3) and not (m3). Sampling depth in lake

Baldegg 40m but 38m on table 2, Lake Geneva 40m but 45 on table 2.

Table 2. see above comment on uncertainties.

Fig S2. Concentrations in Lake Geneva at 310m are much lower throughout the year, varying between 2 and 5 mg/L (Barbier and Quetin 2016). To what year do these profiles correspond?

---

## Author Comment (AC1) · 4 May 2017

Steinsberger et al. investigate the flux of reduced substances (Fred) from sediments in five deep lakes of different trophic states. They found no indication that the trophic state of the lakes controls Fred. However, organic carbon mass accumulation rates together with the mean hypolimnion depth of the lakes relate to Fred and can potentially be used to estimate the influence of Fred to O2 consumption in eutrophic deep lakes. The authors collected a big dataset from five deep lakes and calculate/estimate some factors that are relevant for assessing the hypolimnetic O2 consumption and its driving factors.

The overall topic of the paper falls into the scope of Biogeosciences and it presents some novel and a solid dataset to assess the fluxes of reduced compounds from sediments and its role for hypolimnetic oxygen consumption. The overall presentation of the methods and results/discussion could be improved. The whole manuscript seems lengthy and quite descriptive at many places. I will elaborate on this in more specific comments below.

I would recommend a publication but only after a rewriting of the aims/hypothesis and methods and a restructuring of the discussion to strengthen the main messages.

**Major comments:**

1) The aims of the paper could be reformulated to increase the curiosity of the reader. As it is now, the aims are: extend a dataset/assess constraints/discuss spatial variabilities and consequences: : : This is also reflected in the results/discussion section that is often hard to follow and difficult to say what the authors want to say/conclude here.

**The motivation of this work was the observation in a broad variety of lakes that the areal**

hypolimnetic mineralization rate (AHM) in highly productive lakes could be explained by two components: the O2 consumption at the sediment-water interface, and the O2 consumption by reduced compounds diffusing from the sediment (Müller et al., 2012). In this follow-up project, we focused on the factors that control the fluxes of reduced compounds (methane, ammonium, iron and manganese). Therefore, a large set of porewater fluxes had to be acquired from several lakes with different trophic states, different depths and seasons. Such a laborious work was only possible due to a newly developed method (Torres et al., 2013) allowing on site analyses. The driver for the fluxes of reduced substances was found to be the mass accumulation rate of organic carbon in the sediments. With this heritage from eutrophic times in the sediments we explained the delayed reaction of AHMs in spite of improvements in the TP concentrations of (formerly) eutrophic lakes. In addition, we document that O2 consumption due to reduced substances decreased with mean lake depth, which explains that even in highly productive lakes (such as Lake Geneva) the freshly settling organic matter is well decomposed due to the still elevated deep water O2 concentrations. Thus, TOC mass accumulation rates are small leading to only very small production of reduced compounds in the sediment. As the reviewer states correctly, gathering a large porewater data set and finding a common driver were indeed the aims of this manuscript which we state on P2 L 17-24. In order to improve the manuscript we kindly ask the reviewer to clearly pinpoint the sections

that need improvement.

The first two paragraphs in the "Results and Discussion" (page 5) are only data descriptions without any interpretations jumping from one lake to the other.

We feel that ahead of any interpretation and broader view, the data had to be presented and measurements shown in an illustrative figure. General observations of measured fluxes and specific conditions of lakes have to be presented to guide the reader through the abundance of results. Moreover, our results have to be related and compared with datasets from other studies. Our goal was not yet to interpret each porewater profile from each species. Yet we will modify this paragraph to enhance its structure.

I got easily lost in the details and did not get the major results and their interpretation, something that I would expect at the beginning of this section. I would suggest to reformulate the aims and maybe try to formulate a hypothesis (or hypotheses) or expectations from the data and analyses. With those newly formulated hypotheses the "Results and Discussion" section should be rewritten/-structured, focusing on the new hypotheses.

Working hypotheses and intentions of the study are placed at the end of the introductory section. We are persuaded that measurements, which are the base of all new insights have to be presented at the beginning of the 'Results and Discussion' section so the reader can relate to the subsequent analysis and discussion. Hence, Figure 1 depicts the porewater profiles that are the foundation of all further discussion and conclusions, while Figures 2 and 3 illustrate the main findings of the study.

2) The "Materials and Methods" description has missing information: How many cores were taken per day and depth? I am confused because the authors talk about a "set of cores" collected in Lake Zug (p. 3 line 2).

At each sampling, one core for porewater analysis (p.3 I.5-8), one core for methane analysis (p.3 I. 17) and one core for sediment properties (p.3 I. 22) was taken. One set of cores means one core for porewater analysis, one core for methane analysis and one core for sediment properties was taken. We modified the sentence with to "One set of cores (for porewater analysis, CH4 analysis and bulk sediment parameters) was collected from the permanently oxic part (>4 mgO2 L-1 throughout the year) at 62 m water depth."

Was there any replication or does this refer to the three cores taken for all analyses including reduced substances via capillary electrophoresis, methane and water/TOC content? In the introduction, the authors talk about 50 cores that they took (p. 2 line 21). When I count one core per date and depth for the five lakes (Table 1), I get to 57 cores, which means no replication. How reliable are those data without replication?

There was no replication of the cores as replicating sediment porewater measurements is, at current state of the art, an extremely laborious work which cannot be achieved in a feasible timeframe for the amount of cores collected. In an earlier project (Torres et al., 2013), we compared analyses of porewaters with different methods (CE vs. ion chromatography), which is mentioned in the manuscript. The heterogeneity of the sediment is prone to produce some variability which we acknowledge in the manuscript (p. 5 I. 14-17). Yet our results and analyses are similar to previous measurements e.g Urban et al., (1997), Maerki et al., (2009) and Müller et al., (2012).

And what happened to the 7 cores that do not match with the number stated and my calculations?

A few cores could not be analyzed due to loss of water or damaged core liners, while a few measurement campaigns had to be terminated due to malfunctioning of the CE facility. We now change the number to the exact amount that we use for all calculation (45 cores) and modify the dates accordingly in Table 1.

It would also be nice to read somewhere how many times the reduced substances via capillary electrophoresis, methane and water/TOC content in each core were measured and to what depth.

We agree that this is not entirely clear. Concerning the porewater analysis, we propose to modify the text as follows: "Each porewater sample was analyzed once with two capillary electrophoresis devices each equipped with a capacitively coupled contactless conductivity detector (CE-C4D) (calibrated for anions and cations) directly at the lake shore. (P3 L 10-12) Concerning the methane analysis, we propose to add the following sentence: "The headspace of each CH4 sample was analyzed three times by gas chromatography (Agilent) using a 1010 Supelco Carboxene column with a standard deviation of 0.1 % to 1.3 %." (P3 L 21). The depth of the porewater samples can be seen in Fig. 1 and varied between lakes. The lowest sampling depth was at least 20 cm, which was sufficient to calculate fluxes. All sediment parameters were measured until the lower end of each core, which varied between 30 cm and 55 cm. Yet for the calculation of e.g TOC-MAR we only evaluated the top 10 cm of the sediment.

The distances of the holes are mentioned I could not figure out how deep the sediment was, only from looking at figure 1. From Figure 1, I can also see that it has different numbers (by counting dots) and sometimes different depths and that the distances between points change. This is not mentioned at all in the method description.

The length of sediment cores was between 20 cm to 55 cm, however, this has no effect on the porewater profiles, which are depicted in Figure 1. The spatial resolution of porewater sampling can also be seen from Figure 1, and we think it is not helpful (nor required) to list porewater sampling depths explicitly for all cores. However, we agree that a general statement of the sampling resolution is useful and propose to add the following sentence: "The sampling resolution was 5 mm for the first 5 cm of sediment,  $\leq$  1 cm between 5 cm and 10 cm of sediment,  $\leq$  2 cm between 10 cm and 20 cm of sediment and  $\leq$  3 cm after 20 cm of sediment".

I would like to see a photograph of the cores with the holes, maybe added to the supplement. That would make it much easier to picture such cores.

We see no benefit showing a picture of a PVC tube filled with sediment within this manuscript. Such a picture is presented on the journal cover page of Environmental Sciences – Processes&Impacts Vol. 15/4 (2013) where Torres et al. (2013) was published: (http://pubs.rsc.org/en/content/articlepdf/2013/em/c3em90008h?page=search)

A short description about the literature search in the main text would be helpful. How did the authors search for those data and what did they extract and did they all use similar methods?

We cited the literature that we considered essential, illustrative and supportive for the subject. If we were ignorant about some colleagues' work we are very grateful if you let us know.

3) Assessment of uncertainty of data: The authors provide only limited information on the range of their data. I already asked the question if the authors replicate the sampling at one point on one sampling day and if not how reliable the data are. In figure 1, there are ranges of the data and you can see that especially at the deepest points, there are wide ranges. But in table 2, there is only one value. Did the authors calculate averages for the sampling times? Or are these

data from only one sampling time? It is hard to assess the variability of the data at each sampling point without any knowledge of variation or uncertainty analysis. The authors do not test their results!

The reviewer touches an import topic. As no replicate porewater analyses on more than one core could be made, it is not possible to directly determine the variability of the porewater data. All measurements were carefully performed and the CE instrument was calibrated each time before and during the porewater measurements and checked against the cited multi ion standards with deviations <5% (p.3 I. 15-16). Torres et al. (2013) showed that porewater measurements performed by CE compared to measurements performed by standard ion chromatography yielded similar results. Further, as previously mentioned our data closely agree with earlier studies (Urban et al., (1997), Maerki et al., (2009) and Müller et al., (2012). Therefore, we are confident that the data presented shows state of the art porewater analysis. Local sediment heterogeneity is a matter of constant debate and could not be quantified with the presented experimental investigation. It cannot be decided to what extent the variations in the porewater concentrations were caused by local heterogeneity or temporal variation. We address this issue on p. 5 I. 16-17. We changed Fig. 2 and now show the average Fred values as well as all Fred values measured in the lakes.

Values given in Table 2 show the average flux of a reduced species calculated over all observed values. We will add a sentence to the table to clarify the matter: "Porewater fluxes are averaged over all flux measurements of each individual species".Further we will add the standard deviation of the flux measurements and the Fred calculation for Lake Baldegg and Lake Aegeri. Only for those two lakes we have enough data to justify showing variations. In addition, seasonality of the fluxes is an important issue, however, this goes beyond the scope

In addition, seasonality of the fluxes is an important issue, however, this goes beyond the scope of the present manuscript. It will be treated and discussed in a follow-up modeling paper, which is in preparation.

4) I miss some references throughout the text:

p. 2 line 5: "This relationship suggested a constant fraction of O2 consumption from the sediments, which agreed with the few available estimations from direct of sediment porewater measurements of reduced compounds (ref.)

The entire text from page 1, line 31 to page 2 line 7 refers to work done by Müller et al. (2012a). The reference is cited twice in this paragraph, and we think it is not necessary to cite it a third time after this sentence.

p. 3 line 17-21: a reference for the headspace technique?

In the present literature methane sampling from sediment cores and subsequent analysis from the headspace is described as we do it in this manuscript (e.g. Sobek et al., 2009, Randlett et al., 2015). This is sufficiently clear and comprehensible. We prefer not to refer to other references that describe the same procedure the same way as we do it here.

p. 4 line 26: "The lower TOC-MAR calculation depth of 10 cm was chosen to remain within the timeframe were steady state conditions can be assumed (ref.)."

That is our rational for TOC-MAR calculation in section 2.5. Just recently (after manuscript submission) a paper was published on a related theme by Radbourne et al. (2017). We propose to add a citation to this paper here.

p. 7 line 17-19: "The areal accumulation of TOC per time is controlled by gross sedimentation (which is related to primary production), O2 concentration in the lake bottom water, biological factors like grazing and bioturbation, and physical parameters such as sediment focusing (ref.)." Or is this results taken from the author's own data?

We propose to add here a reference to Sobek et al. (2009), which is already cited at other locations in the text.

5) Why did the authors install the sediment traps at 15 m water depth? All sampling points of the cores are at deeper points and the sedimentation can change with deeper waters, especially because 15 m water depth is above the hypolimnion in most lakes. Does this influence the data and conclusions? Does it play a role and if yes, how? Please also consider discussing this in the main text.

Sediment traps were installed both at 15 m water depth, and at 1 m above the bottom at the deepest location of the lake. The upper traps were used to estimate net export of OC from the productive epilimnion to the deep hypolimnion. They were below the temperature gradient (the metalimnion) at all times.

For the calculation of burial efficiency,only the observations from the lower sediment traps were used and therefore the calculations are independent of the upper sediment trap. We added a sentence in section 2.3 as follows to clarify this: For the calculation of the gross sedimentation only data from the lower trap was used.

Minor points: - P. 2 line 4: : : : : from direct sediment porewater: : : Delete "of"!

**We agree**

- P. 5 line 4: in the four lakes : : : No capital letter!

We propose to change this to it to "the four lakes, Lake Baldegg, Lake ..."

- P. 8 line 3: do you need to say "from the sediments were virtually zero"? Do the authors refer to both lakes that they mention before or only one here?

Yes, this statement refers to both lakes. We propose to change the expression to "close to zero".

- P. 10 line 3: "more commonly available than"

We agree

---

## Author Comment (AC2)

**Reviewer 2**

The paper by Steinsberger et al presents interesting results on the role of dissolved fluxes from sediment in the oxygen consumption in lakes with oxic hypolimnion. It fits perfectly one of the scopes of the journal, linking mainly chemical and physical aspects of the cycle of chemical substances, organic matter, and sedimentation rates. The paper presents new data from fives Swiss lakes with different trophic status. Results, interpretation and conclusion seem coherent, however my main comment concerns i) the lack of clarity in the presentation of the results, and their use in figures. For instance 8 cores were collected in Lake Geneva with corresponding Fred, but only one point plotted (and discussed?) on figure 2 and 3 (average value, deepest point?).

We thank the reviewer for mentioning the problem. We measured the porewater concentrations in eight different cores in Lake Geneva but no cores for TOC measurements nor dating were retrieved. Therefore we relied on the cited data to calculate one average TOC-MAR value for the deep basin of Lake Geneva (P3 Line 31 – P4 line 5). As $F_{red}$ was rather similar at all sampling stations (below >0.1 $gO_2$ $m^{-2}$ $d^{-1}$) and varied only between 0.02 and 0.09 $gO_2$ $m^{-2}$ $d^{-1}$ we decided to plot only one averaged $F_{red}$ value with that average TOC-MAR value. However, we agree with the reviewer that this is unclear and therefore propose to plot all $F_{red}$ values of Lake Geneva into Fig2 and Fig3. But we will not discuss the individual points in the text, as $F_{red}$ was similarly low at all sampling stations.

I have the general feeling that a large set of data has been produced, but partly discussed; and ii) how the variability in the observed fluxes is taken into consideration in the final assessement. On page 5 line 14 and following, the authors correctly indicate that fluxes, at the same location, show variations due to local sediment heterogeneity and/or seasonal effect. Depending on the substances, values varies between 23% to 67%. However, only one value per lake /depth is given in table 2, without any uncertainty, either from the measurements themselves (including uncertainty in sediment accumulation rates) or from the replicates. Then how the values in table 2 are computed (simple average, time weighted)? What could have these uncertainties on the interpretation and conclusion? From a quick evaluation it seems that the main trends are still significant, but this should be discussed in the manuscript to improve the strength of the conclusion.

In table 2, we show average values of all flux measurements at a single sampling station. We agree that it makes sense to present the variability of observations, and therefore propose to add the standard deviations of the flux measurements and $F_{red}$ for Lake Baldegg and Lake Aegeri. Only in these two lakes enough measurements were conducted to justify the calculation of a standard deviation. We also propose to modify the text accordingly. As no duplicate sediment cores were taken, it is not possible to show the variability of the individual flux measurements.

We further propose to add all $F_{red}$ values to Fig.2 to show the encountered variability and to modify the text accordingly. The uncertainties, although considerable, do not change the interpretation or conclusion of the data. At the moment, we are preparing a paper in which we try to explain the encountered seasonal variations with a modelling approach. We believe that incorporating a discussion about the seasonal variations would be beyond the scope of this manuscript and would dilute the main findings of this study.

More detailed comments:

Page 2 line 31. From the classical reference (Wetzel 2001), Lake Geneva is meso-eutrophe (10-30 mg/m3) based on phosphorus content (20 mg/m3), but also on chlorophyll (.

We agree and propose to change this to "meso-eutrophic".

P4 line 23. I don't understand why the sedimentation rate (SR) is calculated based on a depth scale, and then at each layer a TOC-MAR (mass accumulation rate) is calculated, including porosity and dry density. This way is correct if the porosity is relatively constant downcore. But in general in recent sediment porosity vary strongly with depth, and this variation should be taken into account before the computation of the sediment rate. For instance a SR of 2 mm/y correspond to 0.05g cm/2/y with 90%porosity, but 0.1 g/cm2/y with 80% porosity.

We agree that sedimentation rates likely vary downwards. Yet the sedimentation rates over the range of 2-10 cm sediment depth do not change drastically. Based on the characteristic $^{137}$Cs peaks of 1986 and 1963 the sedimentation rate of the top 10 cm can be well established. In Lake Hallwil, no variation in the sedimentation rate over this part of the sediment can be seen. In Lake Aegeri and Lake Baldegg additional to $^{210}$Pb and $^{137}$Cs dating, varve counts over that sediment range were also evaluated and agree well with sedimentation rates previously published (e.g Lotter et. al (1997).
The porosity was calculated for each sediment interval separately with the individual water content and density. The density itself was calculated by the empirical relationship between TOC content and pure geogenic material (Och et. al (2012). We propose to add a section to clarify this and further add the equations for dry density and porosity calculations.

P4 line 24. It is not clearly explain here (but discussed later) why the surface sediments are excluded from the computation.

We explain this in the ensuing sentence P4 line 25 : "The first two centimeters were excluded to neglect freshly deposited matter". We exclude this most of the times very fluffy material, as it possibly reflects just the most recent input to the sediments without any control over long-term deposition to the sediment record. We propose to add the statement : "as this material probably still passes through intense and rapid degradation".

P6 line 28. Not clear here the difference between TOC-MAR and OC (or TOC?) gross sedimentation rate.

We use same nomenclature as the paper cited "OC gross sedimentation rate" by Sobek et. al (2009). We define TOC-MAR on P4 L 21-26 as the organic carbon that is accumulated in the sediments while OC gross sedimentation rate is the deposition rate of OC onto the sediment surface (Sobek et. al 2009) often calculated by sediment trap data (see P6 L29 and Supplement Table S1). We propose to add a sentence to clarify this : "(deposition rate of OC onto the sediment surface)"

P9 line 22. Not clear what is meant by "accessibility of hypolimnetic O2 to the sediment surface"

We mean the $O_2$ flux to the sediments and will change the sentence to : "A closer look on the fluxes of reduced compounds produced by the deposited organic matter in the sediment, however, revealed that they as well depend on the concentration of $O_2$ that the material was exposed to."

Table 1. Units of Hypolimnetic volume is (Mm3) and not (m3).

We will change this to $10^6 \, m^3$.

Sampling depth in Lake Baldegg 40m but 38m on table 2, Lake Geneva 40m but 45 on table 2.

We will make sure that sampling depths are consistent in the revised manuscript.

Fig S2. Concentrations in Lake Geneva at 310m are much lower throughout the year, varying between 2 and 5 mg/L (Barbier and Quetin 2016). To what year do these profiles correspond

We used the most recent data set we had from 2012 from CIPEL. We are now aware that apparently 2012 was one rare year in which $O_2$ levels became high in the deep basin. We will now use data from 2011 which more likely reflect the average $O_2$ concentrations in the deep basin and we will acknowledge CIPEL and INRA for the $O_2$ data of Lake Geneva.